# Evaluation of a multicomponent intervention consisting of education and feedback to reduce benzodiazepine prescriptions by general practitioners: The BENZORED hybrid type 1 cluster randomized controlled trial

**Caterina Vicens**[1,2], **Alfonso Leiva**[2,3]\*, **Ferran Bejarano**[4], **Ermengol Sempere-Verdú**[5], **Raquel María Rodríguez-Rincón**[6], **Francisca Fiol**[1], **Marta Mengual**[4], **Asunción Ajenjo-Navarro**[5], **Fernando Do Pazo**[6], **Catalina Mateu**[1], **Silvia Folch**[4], **Santiago Alegret**[1], **Jose Maria Coll**[7], **María Martín-Rabadán**[8], **Isabel Socias**[2,9]

1 Balearic Health Service IbSalut Son Serra-La Vileta Healthcare Centre, Palma, Illes Balears, Spain, 2 Research Network on Chronicity, Primary Care, and Health Promotion (RICAPPS)-Balearic Islands Health Research Institute (IdISBa), Mallorca, Spain, 3 Balearic Health Service IbSalut, Reseach Unit Primary care Mallorca, Palma, Illes Balears, Spain, 4 Catalan Institute of Health Cat-salut, DAP Camp de Tarragona, Tarragona, Catalunya, Spain, 5 Conselleria de Sanitat Universal i Salut Pública, Paterna Healthcare Centre, Valencia, Comunitat Valenciana, Spain, 6 Balearic Health Service IbSalut Hospital Universitari Son Espases, Pharmacy Department,Palma de Mallorca, Illes Balears, Spain, 7 Balearic Health Service IbSalut, Menorca Primary Care Management, Maó, Illes Baleares, Spain, 8 Balearic Health Service IbSalut, Can Misses Healthcare Centre Ibiza, Illes Baleares, Spain, 9 Balearic Health Service IbSalut, Manacor Healthcare Centre, Manacor, Illes Baleares, Spain

\* aleiva@ibsalut.caib.es

## Abstract

### Background

Current benzodiazepine (BZD) prescription guidelines recommend short-term use to minimize the risk of dependence, cognitive impairment, and falls and fractures. However, many clinicians overprescribe BZDs and chronic use by patients is common. There is limited evidence on the effectiveness of interventions delivered by general practitioners (GPs) on reducing prescriptions and long-term use of BZDs. We aimed to evaluate the effectiveness of a multicomponent intervention for GPs that seeks to reduce BZD prescriptions and the prevalence of long-term users.

### Methods and findings

We conducted a multicenter two-arm, cluster randomized controlled trial in 3 health districts in Spain (primary health centers [PHCs] in Balearic Islands, Catalonia, and Valencian Community) from September 2016 to May 2018. The 81 PHCs were randomly allocated to the intervention group ($n$ = 41; 372 GPs) or the control group ($n$ = 40; 377 GPs). GPs were not blinded to the allocation; however, pharmacists, researchers, and trial statisticians were blinded to the allocation arm. The intervention consisted of a workshop about the appropriate prescribing of BZDs and tapering-off long-term BZD use using a tailored stepped dose reduction with monthly BZD prescription feedback and access to a support web page. The

**Data Availability Statement:** The de-identified data underlying the study's findings are publicly available in the Zenodo repository: https://zenodo.org/record/5607352.

**Funding:** CV received funding from The Carlos III institute from the Ministry of Economy and Competitiveness (grant number PI15/01480). IS received a grant for completing the Doctoral thesis by the Spanish Society of Family and Community Medicine (semFYC.) The funders had no role in study design, data collection and analysis, decision to publish, or preparation of the manuscript.

**Competing interests:** The authors have declared that no competing interests exist.

**Abbreviations:** BZD, benzodiazepine; CFIR, Consolidated Framework Implementation Research; CI, confidence interval; DDD, defined daily dose; GP, general practitioner; IQR, interquartile range; ITT, intention-to-treat; PHC, primary health center.

primary outcome, based on 700 GPs (351 in the control group and 349 in the intervention group), compared changes in BZD prescriptions in defined daily doses (DDDs) per 1,000 inhabitants per day after 12 months. The 2 secondary outcomes were the proportion of long-term users ($\geq$6 months) and the proportion of long-term users over age 65 years.

Intention-to-treat (ITT) analysis was used to assess all clinical outcomes

Forty-nine GPs (21 intervention group and 28 control group) were lost to follow-up. However, all GPs were included in the ITT analysis. After 12 months, there were a statistically significant decline in total BZD prescription in the intervention group compared to the control group (mean difference: −3.24 DDDs per 1,000 inhabitants per day, 95% confidence interval (CI): −4.96, −1.53, $p < 0.001$). The intervention group also had a smaller number of long-term users. The adjusted absolute difference overall was −0.36 (95% CI: −0.55, −0.16, $p > 0.001$), and the adjusted absolute difference in long-term users over age 65 years was −0.87 (95% CI: −1.44, −0.30, $p = 0.003$). A key limitation of this clustered design clinical trial is the imbalance of some baseline characteristics. The control groups have a higher rate of baseline BZD prescription, and more GPs in the intervention group were women, GPs with a doctorate degree, and trainers of GP residents.

## Conclusions

A multicomponent intervention that targeted GPs and included educational meeting, feedback about BZD prescriptions, and a support web page led to a statistically significant reduction of BZD prescriptions and fewer long-term users. Although the effect size was small, the high prevalence of BZD use in the general population suggests that large-scale implementation of this intervention could have positive effects on the health of many patients.

## Trial registration

ISRCTN ISRCTN28272199.

---

## Author summary

### Why was this study done?

- Long-term use of benzodiazepines (BZDs) is common, even though prescription guidelines recommend limiting treatment to a few weeks.

- There is some evidence that interventions implemented by general practitioners (GPs) can reduce long-term prescriptions of BZDs.

- The evidence supporting the effectiveness of these interventions is limited.

### What did the researchers do and find?

- A controlled cluster randomized clinical trial was conducted in 81 primary health centers (PHCs) in 3 regions of Spain to examine the impact of a multicomponent strategy on reducing BZD prescriptions.

- The multicomponent intervention targeted GPs and was based on a 2-hour educational workshop, audit and feedback about prescription practices, and access to a support web page.

- At follow-up, the intervention group had a statistically significant declines in total BZD prescriptions (−3.24 defined daily doses (DDDs) per 1,000 inhabitants per day) and in the percentage of long-term users (−3.6%).

### What do these findings mean?

- Implementing a brief multicomponent intervention that targets GPs can reduce BZD prescriptions.

- A tailored stepped dose reduction for discontinuing BZDs successfully reduced the proportion of long-term users, which could potentially reduce adverse outcomes related to long-term use.

## Introduction

Clinicians mainly prescribe benzodiazepines (BZDs) to treat anxiety and insomnia, or as adjuvants for treatment of depression [1]. Clinical guidelines advocate short-term use of BZDs [2,3] because long-term use leads to tolerance and dependence and is associated with many adverse effects, including somnolence, daytime drowsiness, memory disruption [4–6], increased risk of falls resulting in hip fracture [7–9], and motor vehicle accidents [10,11]. Other studies expressed concerns about possible links of long-term BZD use with mortality [12,13]. Long-term BZD use is particularly inappropriate for older people [14]. However, many clinicians overprescribe BZDs and chronic use is common [15,16]. Overall number of prescriptions of BZDs and BZD-like drugs (Z-drugs) has modestly decreased over the last 10 years in Europe [17]. The prevalence of BZD use varies greatly among countries and ranges from less than 15 defined daily doses (DDDs) per 1,000 inhabitants per day in the United Kingdom, the Netherlands, and Germany, to more than 85 DDDs per 1,000 inhabitants per day in countries such as Iceland, Portugal, and Spain [18].

The Spanish Medicine Agency (AEMPS) reported an average of 87.6 DDDs per 1,000 inhabitants per day during 2018 [19]. According to the most recent Spanish health survey, an average of 10% of the population in Spain reported consuming a BZD during the 2 weeks prior to the survey. Women and aged 65 years or more were the highest consumers (36%) [20].

Most BZDs are mainly prescribed by general practitioners (GPs). The high variability in prescribing BZDs among practices [21] can be explained local health policies regarding the prescription of BZDs, internal conditions of the healthcare practices, and beliefs and attitudes of the GPs regarding the benefits and risks of BZDs [22–24].

Some authors consider that discussing benefits and risk with the patient before the first prescription could be considered a key component in preventing long-term prescriptions of BZDs [22,25]. In addition, a patient who develops dependence may pose a significant challenge to a GP [22,23,26,27]. GPs can use various strategies to gradually taper BZD use in long-term users [28–34].

To change the BZD prescribing behaviors of health professionals and taper BZD use by long-term users, it is important to reduce the risk of dependence and related adverse events from BZD use. The most common deprescribing interventions include the identifying appropriate patients for deprescribing, providing education and development training to the GPs and patients, and using tailored stepped dose reduction of BZDs [35,36]. The most common implementation strategies are targeting professional behavioral changes by using printed educational materials, educational meetings, educational outreach, local opinion leaders, audit and feedback, computerized reminders, and tailored interventions [37–40]. Audit and feedback and educational meetings are widely used in clinical practice as quality improvement measures. Two systematic reviews that examined 220 randomized controlled trials showed these had a small to moderate effect on changing the behaviors of health professionals. However, there was wide variation in their impact and the extent to which they were implemented [40,41].

Some authors suggest that blending of the effectiveness and implementation stages during development of an intervention could improve the translation of the research findings into clinical practice [42,43].

The BENZORED Phase IV trial is a hybrid type 1 effectiveness and implementation study that evaluates the effectiveness and the implementation of an intervention using a GP training workshop on the appropriate use of BZDs. The intervention encourages GPs to gradually taper BZDs for long-term users and provides monthly feedback to GPs about their BZD prescriptions and access to a support web page. This study also aims to identify barriers and facilitators that affect the implementation of this intervention in primary care settings. In the present manuscript, we report the results of the effectiveness outcomes.

## Methods

### Study design

This is a multicenter two-arm cluster randomized controlled type 1 hybrid effectiveness–implementation trial (BENZORED trial) conducted between September 2016 and May 2018. Allocation was performed at the level of the primary health centers (PHCs) to reduce bias due to "treatment contamination." All participant GPs from included PHCs were analyzed, and primary and secondary outcomes were at the GP level. GPs of the PHCs allocated to usual care arm received no intervention.

The details of the protocol were published elsewhere [44] (S1 Study Protocol).

The study protocol was approved by the Primary Care Research Committee, the Balearic Islands Ethical Committee of Clinical Research (IB3065/15), l'IDIAP Jordi Gol Ethical Committee of Clinical Research (PI 15/0148), and Valencia Primary Care Ethical Committee of Clinical Research (P16/024). This study followed the principles outlined in the Declaration of Helsinki (seventh revision).

Only PHCs in which at least two-thirds of the GPs agreed participate were included. Informed consent was waived by the Balearic Islands and the l'IDIAP Jordi Gol Ethical Committee of Clinical Research because the analysis used anonymous administrative data. The requirement for informed consent was not waived by the Valencian Primary Care Ethics Committee, and, therefore, from this region, only GPs who agreed and provided written informed consent were included.

### Study population

PHCs from the following regions of Spain were eligible for participation: Balearic Islands (IbSalut), Catalonia (Institut Català de la Salut; Tarragona-Reus district), and Valencian Community (Conselleria de Salut Universal; Arnau de Vilanova-Llíria district). Healthcare in Spain

is a public service with universal coverage and free access for the entire population. Spanish primary care system consists of 3 organizational levels: the State's central administration agency that is responsible of the general coordination and legislation and the pharmaceutical policy; the delivery of health services are responsible of regions (Autonomous communities) and regions are organized by health districts that include PHCs. PHCs are staffed by multidisciplinary teams comprising of GPs, pediatricians, nurses, gynecologists, and physiotherapists. At baseline, we assessed PHC characteristics (total number of patient listed, proportion of patients aged 65 years or more, urban/rural setting, and training practice) and GP characteristics (age, sex, GPs with 3 years specialty training, doctorate degree, resident trainer, and years working as GP).

## Randomization and masking

A computer-generated random number table was used to allocate the PHCs to a usual care (control) group or a multicomponent intervention group in a ratio of 1:1. All PHCs were randomized at the same time to maintain allocation concealment and were stratified by health districts, PHCs baseline DDDs per 1,000 inhabitants per day, and proportion of patients older than 65 years to ensure PHCs were balanced to those characteristics. GPs were not blinded to the allocation; however, the pharmacists, researchers, and trial statisticians were blinded to the allocation arm. The primary and secondary outcomes were measured using dispensed prescribing data from the electronic clinical records with predefined indicators.

## Intervention

The multicomponent BENZORED intervention consisted of an educational meeting for GPs, audit and feedback, and a support web page.

Educational meetings are commonly used for continuing medical education with the aim of improving professional practice and patient outcomes. Educational meetings include courses, conferences, lectures, workshops, seminars, and symposia [40].

At the start of the trial, the GPs from the intervention group received a 2-hour educational face-to-face workshop in their PHCs. These interventions were delivered by researchers who were GPs, had great expertise in prescribing and deprescribing BZD, and provided training about appropriate procedures for prescribing BZDs. This included educational information about the pharmacological properties of BZDs (biological half-life and equivalent dose of different BZDs), indications for prescription, recommended duration of use, adverse effects, dependence, tolerance, prevalence, and consequences of long-term use. These GPs were asked to discuss with their patients harms and benefits of taking BZD before the first prescription. They received training in a structured BZD discontinuation intervention based on gradual BZD dose reduction and training based on real clinical cases [1,33].

After the initial training workshop, all participating GPs received automated monthly feedback. The audit and feedback, based on the "Feedback Intervention Theory" [45], provided a summary of the clinical performance of healthcare over a specified period of time that was aimed at changing the practices of health professional [41].

Graphical depictions of BZD prescription feedback consisted of a line graph that plotted each GP's monthly BZD prescription rate in DDDs per 1,000 inhabitants per day for 12 months, and 2 additional line graphs that plotted the BZD prescription rate of the PHC and the health district. The GPs also received a password to access to a support web page that provided additional information to reinforce the messages of the workshop. This website included videos, descriptions of practical cases, and an information leaflet for patients about BZDs, Z-drugs, and sleep hygiene (http://benzored.es).

## Outcomes

**Primary outcome measure.** The primary outcome measure was DDDs of BZDs per 1,000 inhabitants per day after 12 months. Total number of dispensed DDD from the N05BA (diazepam, chlordiazepoxide, potassium clorazepate, lorazepam, bromazepam, clobazam, ketazolam, alprazolam, halazepam, pinazepam, clotiazepam, bentazepam), N05CD (flurazepam, flunitrazepam, triazolam, lormetazepam, midazolam, brotizolam, quazepam, lorprazolam) and N05CF (zopiclone, zolpidem, zaleplon) groups of the Anatomical Therapeutic Chemical (ATC) Classification System code were extracted from the e-prescription databases of each health district. Total number of dispensed DDD was divided by total GP's patient list / 1,000 × 365 days.

**Secondary outcome measures.** One secondary outcome measure was the proportion of all patients who were long-term BZD users after 12 months.

The other secondary outcome measure was the proportion of long-term BZD users who were aged 65 years or more after 12 months.

Long-term use was defined as a continuous prescription for any dose of BZD for a minimum of 6 months.

The RELE (IbSalut, Balearic Islands), Rec@p (Institut Català de la Salut; Tarragona-Reus district), and Receta electrónica (Conselleria de Salut Universal; Arnau de Vilanova-Llíria district) are the e-prescription systems and databases used in the 3 health districts included in the study. These databases contain information about all primary care prescription dispensed in community pharmacies, including dispensing date, pharmaceutical product (active ingredient and brand name), dose, and treatment duration.

## Data management

**Statistical analysis.** The sample size calculation was based on the DDDs per 1,000 inhabitants per day. The mean DDDs per 1,000 inhabitants per day was 89.3 DDDs, the standard deviation was 18.7, and the intracluster correlation coefficient was 0.05 [46]. To detect a clinically meaningful difference of 5 DDDs per 1,000 inhabitants per day between the intervention and control groups, with 80% power and a two-sided α value of 0.05, at least 64 PHCs with an average of 10 GPs per PHC were needed. Adjustment for clustering was performed by calculation of $1 + (n - 1) \times \rho$, where n is the average cluster size and ρ is the intracluster correlation coefficient [46]. For the primary outcome, the effectiveness of the intervention was analyzed using a mixed effects Poisson regression model to account for clustering at the level of the PHC and adjusted for baseline GPs DDDs per 1,000 inhabitants per day. Also, adjusted average marginal effects for group were calculated from the mixed effects Poisson regression models. For secondary outcomes, a random effect Tobit regression with 2 censored values at 0 and 100 was also carried out to analyze the effectiveness of the intervention in the proportion of long-term BZD users after 12 months and proportion of long-term BZD users after 12 months over 65 years old adjusted for GPs baseline proportion of long-term BZD users.

Missing outcomes were accounted for using multiple imputation with chained equation [47] to estimate DDDs per 1,000 inhabitants per day from GPs lost to follow-up. Fifty imputed samples were generated, and estimates were combined using Rubin rules. Baseline values and PHC were included in the imputation model. All statistical analysis were performed using SPSS v.23.0 (IBM, Armonk, NY, USA) using a predefined analysis plan (S1 Analysis plan). Intention-to-treat (ITT) analysis was used to assess all clinical outcomes, and the results are reported in accordance with the Consolidated Standards of Reporting Trials guidelines extension for cluster trials (S1 CONSORT Checklist) [48] and the Template for intervention Description and Replication (S1 TIDieR Checklist).

**Subgroup analysis.** To assess whether the effect of the intervention on BZD prescription varied according to GP characteristics (sex and working years), characteristic of the GP's practice list (percentage of patient aged 65 years old and sex) and health districts, a subgroup analysis of effectiveness that was previously planned in the protocol was performed.

**Implementation process.** Full details of the methods used for evaluation of the implementation of the BENZORED IV intervention were reported elsewhere [49]. To summarize, 40 semistructured interviews and 5 focus group meetings were conducted. GPs were invited to participate in an effort to ensure representativeness for the different health district locations, and they were classified as "low prescribers" or "high prescribers" based on the 12-month final evaluation.

The Consolidated Framework Implementation Research (CFIR) was used to guide development of the focus group meetings and for coding and data analysis [42]. The CFIR is a theoretical framework that provides a list of 41 constructs organized in 5 domains that can negatively or positively influence implementation. Among the 45 GPs contacted, 40 agreed to interviews.

The implementation fidelity was measured as adherence to the intervention, defined as whether "a program service or intervention is being delivered as it was designed" [50]. Adherence was measured using a questionnaire given to all GPS in the intervention group (S1 Questionnaire). This questionnaire asked the GPs to rank adherence to each of the following components of the intervention on a scale from 1 to 10: discussing benefits and risk with the patient before the first prescription; setting limits for the duration from the start of treatment; tailoring stepped dose reduction to discontinue BZD use in long-term users; reviewing BZD prescription feedback; downloading and using the patient information leaflet about BZDs, Z-drugs, and sleep hygiene; and visiting the web page to reinforce information.

## Results

### Enrollment and disposition of PHCs and GPs

This study was conducted in 3 Spanish health districts, which included 90 PHCs (58 in the Balearic Islands, 20 in the Tarragona-Reus district, and 12 in the Arnau de Vilanova-Llíria district) and 867 eligible GPs (Fig 1). Eighty-one PHCs (90%) agree to participate and were randomly allocated 41 to the intervention arm and 40 to the usual care arm. Overall, there were 749 GPs (86.4%), with 482 from the Balearic Islands, 177 from the Tarragona-Reus district, and 90 from the Arnau de Vilanova-Llíria district. A total of 372 GPs were in the intervention arm and 377 were in the usual care arm. During follow-up, 49 GPs were excluded due to changes in workplace. However, all GPs were included in the ITT analysis.

Comparison of the 2 groups at baseline indicated that PHCs in the intervention arm and usual care arm were similar in registered patient population size and percentage of patients aged 65 years or more (Table 1). However, the average DDDs per 1,000 inhabitants per day was slightly higher in the usual care group, 74.6 in the intervention group and 76.9 in the control group, and more GPs in the intervention group were women, GPs with a doctorate degree, and trainers of GP residents.

### Primary outcome

After 12 months, the number of BZD prescriptions (DDDs per 1,000 inhabitants per day) in the intervention group decreased from 74.6 (95% confidence interval [CI]: 71.4, 77.8) to 71.0 (95% CI: 67.8, 74.1) (Table 2). The corresponding decrease in the control group was from 76.9 (95% CI: 73.6, 80.3) to 76.5 (95% CI: 73.2, 79.8). The between groups difference adjusted by baseline BZD prescription in total BZD prescriptions at 12 months (DDDs per 1,000 inhabitants per day) was −3.24 DDDs per 1,000 inhabitants per day (95% CI: −4.96, −1.53, $p < 0.001$). At the PHC level, the intracluster correlation coefficient of DDDs per 1,000

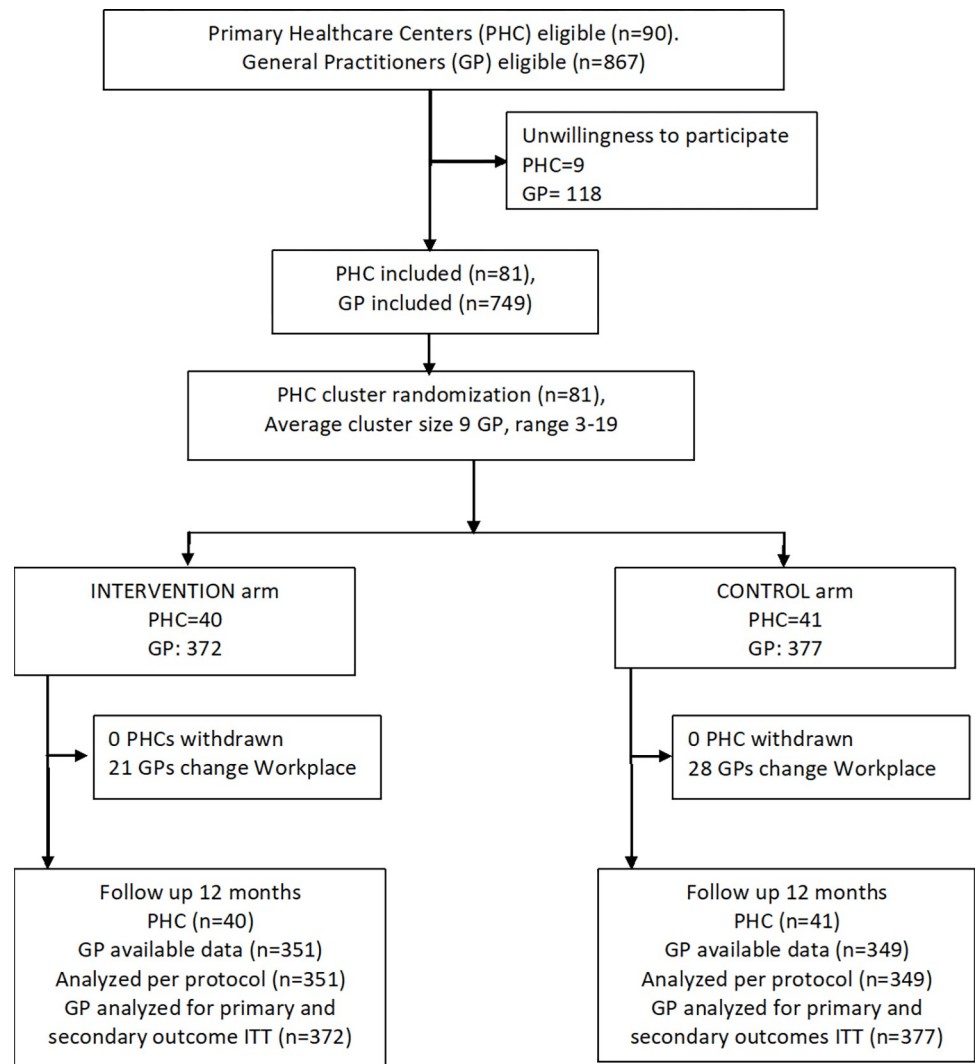

**Fig 1. Flow chart of the study.** GP, general practitioner; ITT, intention-to-treat; PHC, primary health center.

inhabitants per day of BZDs was 0.30. This was higher than expected from the sample size calculation, indicating substantial within-PHC clustering for BZD prescriptions.

We performed subgroup analysis by health districts. The results indicated some differences in the effectiveness of the intervention, the interaction term from DDDs per 1,000 inhabitants per day and group was statistically significant among health districts ($p = 0.002$). We reported effect sizes by health district in Table 1 of the Supporting information (S1 Table).

The between groups differences adjusted by baseline BZD prescription in the total DDDs per 1,000 inhabitants per day in the Tarragona-Reus was −6.8 (95% CI −12.2 to −1.30, $p = 0.023$), in the Arnau de Vilanova llíria was −3.3 (95% CI −7.5 to 1.2, $p = 0.156$) and in the Balearic Islands (IbSalut) was −2.8 (95% CI −4.3 to −1.3, $p < 0.001$).

**Table 1. Distribution of PHCs and GPs' characteristics at baseline.**

|  | Intervention n/N (%) | Control n/N (%) |
|---|---|---|
| PHC characteristics | N = 40 | N = 41 |
| Baseline DDD per 1,000 inhabitants per day | 74.6 ± 31.6 | 76.9 ± 33.1 |
| Total number of patients listed |  |  |
| <12,500 | 10/40 (25.0) | 11/41 (26.8) |
| 12,500–25,000 | 21/40 (52.5) | 21/41 (51.2) |
| >25,000 | 9/40 (22,5) | 9/41 (22,0) |
| Proportion of patients > = 65 years |  |  |
| Mean ± SD | 16.3 ± 3.1 | 16.1 ± 3.6 |
| Urban centers | 26/40 (65.0) | 28/40 (70.0) |
| Training PHC | 16/40 (40.0) | 14/40 (35.0) |
| GP characteristic | N = 372 | N = 377 |
| Age (years) |  |  |
| Mean ± SD | 51.7 ± 8.8 | 51.9 ± 8.6 |
| Women | 222/372 (59.7) | 206/377 (54.6) |
| GP with 3 years specialty training | 324/372 (87.1) | 328/377 (87.0) |
| PhD (doctorate degree) | 36/372 (9.7) | 23/377 (6.1) |
| GP resident trainer | 92/372 (24.7) | 70/377 (18.6) |
| Years working as GP |  |  |
| Mean ± SD | 21.9 ± 9.6 | 21.6 ± 9.7 |
| Years working as GP in the actual workplace |  |  |
| Mean ± SD | 11.5 ± 8.7 | 10.7 ± 9.5 |

DDD, defined daily dose; GP, general practitioner; PHC, primary health center; SD, standard deviation.

## Secondary outcomes

The intervention and control groups had statistically significant differences in both secondary outcome measures (Table 2). In particular, the adjusted by baseline BZD prescription, difference of long-term BZD users among all patients was −0.36 percentage points (95% CI: −0.55, −0.16, $p < 0.001$) and the corresponding adjusted relative reduction was 3.6% (95% CI: 1.65, 8.86, $p < .001$). In patients aged 65 years or more, the adjusted difference was −0.87 percentage points (95% CI: −1.44, −0.30, $p = 0.003$) and the corresponding adjusted relative reduction was 3.5% (95% CI: 1.19, 5.71, $p = 0.004$).

## Evaluating the implementation process

A total of 246/372 (66%) GPs from the intervention group answered this questionnaire. These GPs generally valued the multicomponent intervention as flexible and integrable into their practices. However, the 3 components of the intervention were valued differently. They reported the educational workshop was useful for clinical practice and the BZD prescription feedback was easy to interpret. They reported the tailored stepped dose reduction regimen for discontinuing BZDs was complex, and their lack of time and workload were significant barriers to implementation. The material included in the web page was considered helpful for the patients, but they reported some technical problems in accessing the website.

The fidelity of adherence to the intervention differed among the different components. Discussing benefits and risk with the patient before the first prescription had a median adherence score of 8 (interquartile range [IQR]: 7 to 9); setting limits for the duration from the start of the treatment had a median score of 8 (IQR: 7 to 9); tailoring stepped dose reduction to

**Table 2. Comparison of GPs' BZD prescriptions, percentage of all BZD long-term users (≥6 months) and percentage of BZD long-term users aged 65 or more.**
Unadjusted, adjusted (DDD per 1,000 inhabitants per day and percentage of BZD long-term users at baseline) and estimated ITT results for the control and intervention groups at 12-month follow-up.

| | Intervention Mean ± SD | Control Mean ± SD | Total | | | | | |
| --- | --- | --- | --- | --- | --- | --- | --- | --- |
| | | | Unadjusted mean difference (95% CI) | *p*-value | Adjusted mean difference (95% CI) | *p*-value | Estimated ITT mean difference (95% CI) | *p*-Value |
| **Primary Outcome** | | | | | | | | |
| DDD per 1,000 inhabitants per day at baseline | 74.6 ± 31.6 | 76.9 ± 33.1 | | | | | | |
| DDD per 1,000 inhabitants per day at 12 months | 71.0 ± 29.9 | 76.5 ± 31.5 | −5.38 (−9.94; −0.84) | 0.020 | −3.45 (−5.09; −1.82) | <0.001 | −3.24 (−4.96; −1.53) | <0.001 |
| **Secondary outcome** | | | | | | | | |
| Percentage of BZD long-term users at baseline | 9.5 ± 3.9 | 9.8 ± 3.9 | | | | | | |
| Percentage of BZD long-term users at 12 months | 9.1 ± 3.6 | 9.8 ± 3.8 | −0.68 (−1.23; −0.12) | 0.016 | −0.38 (−0.57; −0.18) | <0.001 | −0.36 (−0.55; −0.16) | <0.001 |
| Percentage of BZD long-term users >65 years old at baseline | 24.9 ± 8.8 | 25.3 ± 8.2 | | | | | | |
| Percentage of BZD long-term users >65 years old at 12 months | 23.9 ± 7.7 | 25.2 ± 7.9 | −1.28 (−2.45; −0.12) | 0.031 | −0.90 (−1.46; −0.34) | 0.002 | −0.87 (−1.44; −0.30) | 0.003 |

BZD, benzodiacepine; CI, confidence interval; DDD, defined daily dose; GP, general practitioner; ITT, intention-to-treat; SD, standard deviation.

discontinuing BZD use in long-term users had a median score of 7 (IQR: 7 to 8); reviewing BZD prescription feedback had a median score of 7 (IQR: 6 to 8); downloading and using the patient information leaflet about BZDs, Z-drugs, and sleep hygiene had a median score of 7 (IQR: 7 to 8); and visiting the web page to reinforce information had a median score of 5 (IQR: 3.75 to 7).

## Discussion

Our major finding is that a multicomponent intervention that targeted GPs was effective in reducing prescriptions of BZDs and in reducing the proportion of long-term BZD users at 12 months.

### Comparison with existing literature

Our study indicated a small reduction in the total number of BZD prescriptions. A previous study found greater reductions in the prescriptions of BZDs to elderly patients (measured as DDD per 1,000 inhabitants per day) following continuous educational outreach visits to GPs. This previous study used a more intensive intervention that consisted of visits between GPs and pharmacists every 2 to 8 weeks [51]. However, these more intensive interventions are more difficult to implement in most PHCs. In contrast, other authors found that educational visits with GPs regarding BZD prescriptions were not effective in reducing the number of prescriptions [52,53]. Pimlott and colleagues and Holm also found that an educational intervention and feedback to GPs was not effective [54,55]. However, these studies have some methodological limitations.

Changing inappropriate health interventions is essential for minimizing patient harm, maximizing efficient use of resources, and improving population health. Prescribing BZDs in patients aged 65 years or more is considered potentially inappropriate and should be avoided because the decreased metabolism of long-acting agents and increased sensitivity and risk of

cognitive impairment, delirium, falls, fractures, and motor vehicle crashes in older adults [14]. We found that our intervention led to a 3.5% reduction in the proportion of long-term BZDs users who were aged 65 years or more. There is evidence that certain interventions delivered by GPs were effective in reducing the number of long-term BZD users [27]. A previous study of a stepped dose reduction of BZDs with written instructions delivered by GPs to long-term users showed that the intervention was 3 times more effective than usual care [33]. Similar results (25% reduction of BZD use by long-term users) were reported in an intervention consisting of sending an educational booklet and a component for risk self-assessment of BZD use to long-term users [56]. These results differ from our finding that the intervention implemented by GPs provided only a modest reduction in the percentage of long-term BZD users. In more pragmatic clinical trials, GPs receive educational training, and all patients are evaluated, but not all patients receive the intervention. However, a small effect in a pragmatic trial that includes most GPs and all eligible patients is clinically relevant in absolute terms, because overuse of BZDs is widespread in Spain.

The overall reduction in BZD prescriptions that we observed in the intervention group was similar in magnitude to that found in an intervention that included education with audit and feedback [40,41].

Our results indicated that the intervention led to a reduction of 3.2 DDD per 1,000 inhabitants per day. From a health system perspective, this reduction over a 1-year period is clinically important. Some other health systems are steadily decreasing their prescriptions for BZDs. For example, Norway decreased BZD prescriptions from 65.7 to 49.7 DDD per 1,000 inhabitants per day from 2008 to 2018. The decline in Denmark over the same period was 42.4 to 25.9 DDD per 1,000 inhabitants per day. However, such decreases did not occur in Spain, Portugal, or France. In fact, over this same time, Spain had an increased consumption of BZDs from 75.5 to 89.3 DDD per 1,000 inhabitants per day [18].

## Strengths and limitations

To our knowledge, this is the largest clinical trial to analyze the effectiveness of an educational intervention for GPs in a primary care setting that focused on reducing prescriptions for BZDs. We included all GPs of the participating PHCs. The study had a high rate of GP participation compared to similar studies [54], and very few GPs were lost to follow-up. We also found differences in effect size among the different health districts, indicating that the characteristics of the health district affected the efficacy of the intervention. For example, the Balearic Islands and Tarragona have local policies for GPs regarding BZD prescribing that include indicators and incentives to motivate GPs to reduce their BZD prescription. Another strength of our study is that it was performed in 3 health districts that have different characteristics. This suggests the results may be applicable in different clinical settings.

Our study had also some limitations. At baseline, the intervention group had more GP training healthcare centers and more GP trainers and physician doctors. This could have led to an overestimate of the effectiveness of the intervention because GP trainers and physician doctors are more likely to follow clinical guidelines and tend to write fewer BZD prescriptions [21]. Although we did not adjust for GP characteristics, we did adjust for baseline BZD prescriptions to minimize the potential of selection bias. GPs in the intervention PHCs may have shared information and strategies with GPs of the usual care PHCs. However, we randomized PHCs to avoid cross-contamination, and the PHCs were mostly in different villages or cities. Moreover, only GPs in the intervention arms had access to the additional material from the website and received individual BZD prescription feedback that was sent to their institutional e-mail addresses.

### Implications for clinical practice and future research

A minimal intervention provided to GPs in a primary care setting, which consists of an educational training program and feedback regarding BZD prescriptions, may help to reduce the overall number of BZD prescriptions and reduce the number of long-term BZD users. From a public health perspective, because BZD use is very common in Spain, a small effect on long-term use by individuals could have a large impact in preventing adverse effects related to BZD use at the level of the population, such as falls, fractures, or cognitive impairment, especially in the elderly. In conclusion, our rigorous trial design and theory-based evaluation of the implementation process provided evidence of the effectiveness of a multicomponent strategy that targeted GPs to help reduce BZD prescriptions and the number of long-term users.

## Supporting information

**S1 CONSORT Checklist. Consolidated Standards of Reporting Trials guidelines extension for cluster trials checklist.**
(DOC)

**S1 StaRI Checklist. Standards for Reporting Implementation Studies: the StaRI checklist.**
(DOCX)

**S1 TIDieR Checklist. Template for Intervention Description and Replication checklist.**
(DOCX)

**S1 Table. Subgroup analysis of effectiveness of the intervention at 12-month follow-up by health districts.**
(DOCX)

**S1 Study Protocol. BENZORED protocol.**
(DOC)

**S1 Analysis plan. Predefined analysis plan and amendments.**
(DOCX)

**S1 Questionnaire. Implementation Fidelity Questionnaire.** Six items fidelity questionnaire.
(PDF)

## Acknowledgments

We are grateful to the participating GPs and the Heads of the PHCs for their help with project development.

## Author Contributions

**Conceptualization:** Caterina Vicens, Alfonso Leiva, Ermengol Sempere-Verdú, Francisca Fiol, Isabel Socias.

**Data curation:** Marta Mengual, Asunción Ajenjo-Navarro, Fernando Do Pazo, Silvia Folch, Jose Maria Coll, María Martín-Rabadán.

**Formal analysis:** Alfonso Leiva, Raquel María Rodríguez-Rincón, Fernando Do Pazo, Silvia Folch.

**Funding acquisition:** Caterina Vicens, Ferran Bejarano, Ermengol Sempere-Verdú, Santiago Alegret, Jose Maria Coll.

**Investigation:** Caterina Vicens, Ferran Bejarano, Ermengol Sempere-Verdú, Raquel María Rodríguez-Rincón.

**Methodology:** Alfonso Leiva.

**Project administration:** Ermengol Sempere-Verdú, Catalina Mateu.

**Resources:** Ferran Bejarano, Francisca Fiol, Marta Mengual, María Martín-Rabadán, Isabel Socias.

**Supervision:** Raquel María Rodríguez-Rincón, Francisca Fiol, Asunción Ajenjo-Navarro, Fernando Do Pazo, Catalina Mateu, Santiago Alegret, Jose Maria Coll, María Martín-Rabadán.

**Validation:** Raquel María Rodríguez-Rincón, Marta Mengual, Asunción Ajenjo-Navarro, Fernando Do Pazo, Silvia Folch, Santiago Alegret, Jose Maria Coll.

**Writing – original draft:** Caterina Vicens, Alfonso Leiva, Ferran Bejarano, Ermengol Sempere-Verdú, Isabel Socias.

**Writing – review & editing:** Raquel María Rodríguez-Rincón, Francisca Fiol, Marta Mengual, Asunción Ajenjo-Navarro, Fernando Do Pazo, Catalina Mateu, Silvia Folch, Santiago Alegret, Jose Maria Coll, María Martín-Rabadán.

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
