## [Editor Report · Decision Letter 0]

25 Aug 2021

Dear Dr Leiva Rus, 

Thank you for submitting your manuscript entitled "Effectiveness of a multicomponent intervention consisting of education and feedback on reducing benzodiazepine prescriptions by general practitioners: BENZORED hybrid type I cluster randomized controlled trial" for consideration by PLOS Medicine.

Your manuscript has now been evaluated by the PLOS Medicine editorial staff and I am writing to let you know that we would like to send your submission out for external peer review.

Please re-submit your manuscript within two working days, i.e. by Aug 27 2021 11:59PM.

Kind regards,

Caitlin Moyer, Ph.D.

Associate Editor

PLOS Medicine

---

## [Decision Letter · Decision Letter 1]

21 Oct 2021

Dear Dr. Leiva,

Thank you very much for submitting your manuscript "Effectiveness of a multicomponent intervention consisting of education and feedback on reducing benzodiazepine prescriptions by general practitioners: BENZORED hybrid type I cluster randomized controlled trial" (PMEDICINE-D-21-03642R1) for consideration at PLOS Medicine. 

Your paper was evaluated by a senior editor and discussed among all the editors here. It was also discussed with an academic editor with relevant expertise, and sent to three independent reviewers, including a statistical reviewer. The reviews are appended at the bottom of this email and any accompanying reviewer attachments can be seen via the link below:

[LINK]

In light of these reviews, I am afraid that we will not be able to accept the manuscript for publication in the journal in its current form, but we would like to consider a revised version that addresses the reviewers' and editors' comments. Obviously we cannot make any decision about publication until we have seen the revised manuscript and your response, and we plan to seek re-review by one or more of the reviewers. 

We expect to receive your revised manuscript by Nov 11 2021 11:59PM. Please email us (plosmedicine@plos.org) if you have any questions or concerns.

We look forward to receiving your revised manuscript. 

Sincerely,

Caitlin Moyer, Ph.D.

Associate Editor

PLOS Medicine

plosmedicine.org

From the Academic Editor:

1. Please include a description of the intervention, using the TIDIER checklist or equivalent as a guide.

2. Please include the CONSORT checklist for cluster randomized trials.

3. Please describe the intra-cluster correlation coefficient in the Results section and please clarify if this was lower than originally assumed for the sample size calculation.

Other editorial points:

4. Data availability statement: PLOS Medicine requires that the de-identified data underlying the specific results in a published article be made available, without restrictions on access, in a public repository or as Supporting Information at the time of article publication, provided it is legal and ethical to do so. Please see the policy at

http://journals.plos.org/plosmedicine/s/data-availability

and FAQs at

http://journals.plos.org/plosmedicine/s/data-availability#loc-faqs-for-data-policy

Thank you for your willingness to make your data available in a repository. However, the file “repository.sav” appears to be a proprietary format (SPSS dataset) that may reduce the accessibility. We request that authors provide data in file formats that are standard in their field and allow wide dissemination. If there are currently no standards in the field, authors should maximize the accessibility and reusability of the data by selecting a file format from which data can be efficiently extracted.

5. Throughout text: Please include line numbers with the revised version.

6. Abstract: Please report your abstract according to CONSORT for abstracts, following the PLOS Medicine abstract structure (Background, Methods and Findings, Conclusions) http://www.consort-statement.org/extensions?ContentWidgetId=562

7. Abstract: Please specify who was blinded to the intervention and control, and please mention any GPs or PHCs that were lost to follow up in each group.

8. Abstract: Methods and Findings: Please quantify the main results for decline in BZD prescriptions and proportions of long term users over age 65 including both 95% CIs and p values. Please present findings for long term users in general.

9. Abstract: Please report a summary of any adverse events in the Abstract (and please also present these findings in the Results of the manuscript).

10. Abstract: In the last sentence of the Abstract Methods and Findings section, please describe the main limitation(s) of the study's methodology.

11. Author summary: At this stage, we ask that you include a short, non-technical Author Summary of your research to make findings accessible to a wide audience that includes both scientists and non-scientists. The Author Summary should immediately follow the Abstract in your revised manuscript. This text is subject to editorial change and should be distinct from the scientific abstract. Please see our author guidelines for more information: https://journals.plos.org/plosmedicine/s/revising-your-manuscript#loc-author-summary

12. Main text: For citations within the text, please include the reference number within square brackets, placed before the sentence punctuation. Where multiple references are indicated, please do not include spaces within the brackets.

13. Methods: Please include the study protocol/predefined analysis plan, with any amendments, as Supporting Information.

14. Methods: Please describe the nature of GP consent to participate.

15. Methods: Study population: Please mention the three regions are part of Spain.

16. Methods: Please complete the CONSORT checklist and ensure that all components of CONSORT are present in the manuscript.

In addition to the CONSORT checklist, we suggest that your implementation research be reported according to Standards for Reporting Implementation Studies statement (STARI). The STARI guidelines can be found here: https://www.equator-network.org/reporting-guidelines/stari-statement/

17. Methods: Please describe the intervention (educational workshop, training on discontinuation/dose reduction intervention, automated monthly feedback, support web page) in greater detail.

18. Methods: Please provide more description of how the DDD per 1000 inhabitants per day was calculated.

19. Methods: Please provide more description of the prescriptions claims databases used to extract prescription records.

20. Methods: Please provide additional description of the subgroup analyses.

21. Methods: The trial registration lists the outcomes of “3. Feasibility, adoption and fidelity of the intervention will be measured by an "ad hoc" questionnaire to measure GP opinion”. (a) Can you please present those results as part of this manuscript, or indicate why that is not possible? (b) Can you please indicate when you plan to publish those results?

22. Methods: Please provide more detail on the implementation aspect of the trial. Please clearly define and differentiate and define your intervention and implementation strategy and outcomes for each. It should be clear how both aspects were tested for this hybrid trial. As described, the multicomponent interventions seem more like implementation strategies.

Please consider using the guidelines published by Proctor et al. to guide your reporting: Proctor EK, Powell BJ, McMillen JC: Implementation strategies: recommendations for specifying and reporting. Implement Sci 2013, 8:139

Please ensure you have included relevant and sensitive implementation outcomes to capture the impact of individual, team-level, and organizational factors that could determine the success/failure of the intervention. Example document on Implementation outcomes. Please use standard terminology for implementation outcomes so that there is consistency in reporting (e.g. adoption, fidelity, penetration, sustainability etc.).

Please clarify whether you used suitable implementation theoretical frameworks to guide the design of the study, analysis, and interpretation of your findings.

23. Results: The sample size listed in the submitted manuscript (749 GPs) and the target from the trial registry (508 GPs) differ. Please explain the discrepancy.

24. Results: Please provide 95% CIs and p values for all primary and secondary outcomes and subgroup analyses described in the text. For the subgroup analyses, please report the interaction between health district and outcome. When reporting adjusted analyses, please also mention the factors adjusted for.

25. Results: Please report any findings pertaining to adverse events for the study including numbers of specific events and whether or not adverse events are thought to be related to the intervention.

26. Discussion: Please present and organize the Discussion as follows: a short, clear summary of the article's findings; what the study adds to existing research and where and why the results may differ from previous research; strengths and limitations of the study; implications and next steps for research, clinical practice, and/or public policy; one-paragraph conclusion.

27. References: Please use the "Vancouver" style for reference formatting, and see our website for other reference guidelines https://journals.plos.org/plosmedicine/s/submission-guidelines#loc-references

28. Tables 1 and 2: Please include these in the main text of the manuscript.

29. Table 2: Please note in the legend factors adjusted for in the analyses. Please also include results of unadjusted analyses.

30. Supporting Information Table 1: Please include p values for the subgroup analyses by health district, including the analysis for the interaction, for effectiveness of intervention within districts and comparisons between health districts.

Comments from the reviewers:

Reviewer #1: Review

In short: relevant topic, large well-conducted study, conventional intervention, concise presentation.

Abstract

I would recommend to describe the effect size as small, similar to the conclusion in the discussion section.

Introduction

It would be relevant to specify that the prescription rates for BZB in Spain are relatively high compared to other countries (information that I got from the discussion).

The choice of the implementation strategies (education, feedback, and information platform) should be founded in preparatory research and/or conceptual analysis of current practice. Currently it remains unclear why exactly these strategies were chosen, which is not consistent with current implementation science.

Deprescribing (and de-implementation generally) has received considerable interest in recent years. However, this manuscript does not relate to the implementation science literature on the topic. 

The chosen approach is conventional for the implementation of recommended practices, although well-conducted at large scale. The authors argue that the prevention of first prescription of BZB has rarely been studied. This may strictly be true, but not prescribing has certainly be a topic in other medication.

Methods

The study is described as Type 1 trial, which implies a primary outcome that relates to clinical or health outcomes. The primary outcome is a medication prescription rate, which seems to me as aspect of physician behaviour, thus Type 3 hybrid effectiveness-implementation trial.

The stratification component of the randomization is not quite clear, because it seems to relate to geographical districts rather than practices.

I would recommend to describe the implementation strategies (education, feedback, info-platform) in more detail, using a reporting guideline such as TIDIER as appendix to the manuscript.

The number of outcomes in this manuscript is 3, which is rather. It seems that information on intervention fidelity is not available or not reported, but would be helpful to interpret the findings. 

Results

A participation rate of 90% of invited practices is very high compared to many other studies in primary care. The authors may discuss this (in the discussion section) .

While the study is large and seems well-conducted, the manuscript does not present many results.

Discussion

Similar to the introduction section, the discussion could relate more to the implementation science literature, e.g. Cochrane review on audit and feedback and literature on de-implementation (stopping practices) in healthcare.

Michel Wensing

Reviewer #2: Alex McConnachie, Statistical Review

Vicens et al presents a report of BENZORED, a cluster randomised trial of an educational and feedback intervention for GPs to reduce benzodiazepine prescriptions. This review looks at the use of statistics in the paper.

Overall, I found this to be a nicely written paper. The statistical methods are generally good, with appropriate methods used to account for the cluster randomised design. I do have a few comments, which are generally minor, and should not affect the underlying message of the paper.

Much is made of the "Intention to Treat" analysis. This appears to have been interpreted as an analysis with multiple imputation to account for missing outcome data for some GPs. For me, this is not what ITT means. As the term implies, ITT is about analysing data from a randomised trial according to the intervention that was intended to be delivered. I.e., analysis according to the randomised allocation, regardless of whether the intervention was delivered as planned. In an RCT, all participants should be followed up, even those who withdraw from the intervention.

Loss to follow-up in randomised trials is common, but has nothing to do with the concept of ITT. In my opinion, the primary analysis should be limited to those GPs with follow-up data. This can still be labelled as "ITT", so long as the GPs are analysed according to their original randomised allocation. The amount of missing follow-up data should be reported, and the reader can judge the validity of the primary analysis in the context of this missing data.

As a sensitivity analysis, to assess the impact of missing data on the primary analysis result, multiple imputation is good method to use. However, I always feel that including the randomised group in the imputation process makes this circular; if the outcomes are imputed based on the intervention effect observed in those with complete data, this will tend to reinforce the complete case analysis. I would rather see the multiple imputation carried out without randomised group as a predictor, to see whether the primary analysis result is robust to the assumption of no intervention effect in those with missing outcomes.

That is probably my main point, and is one of emphasis more than anything. Looking at the tables, the complete case analysis appears to give similar results to what is currently described as "ITT". Given the relatively good follow-up achieved in the study, I would not expect multiple imputation without use of randomised group as a predictor to eliminate the intervention effect.

My others points are more minor.

The background reports a figure of 87.6 DDDs per 1000 patients, and a figure of 8.76% of patients taking a DDD each day. These are two different things, so if this is true, then it is quite a coincidence. E.g. 87.6 DDDs per 100 patients could represent about 175 in every 1000 patients (17.5%) all consuming half a DDD each day, or 4.4% of patients taking 2 DDDs every day.

I think the paper would benefit from a clearer statement about the unit of randomisation (PHC) and the unit of analysis (GP). It becomes clear as you go through the paper, but is not immediately obvious. For example, the sample size section talks about data aggregated at the cluster level (PHC), but then makes an assumption about the ICC, which implies analysis of data by GP, clustered within PHCs.

The primary analyses uses Poisson regression, but the intervention effects are reported as mean differences. How were the regression results converted to absolute differences?

There is mention of the trial being reported according to the CONSORT guideline for cluster trials, but I did not see a checklist. One feature that is missing from the results is a report of the ICCs observed. This would be useful for future researchers.

I do not think it is important to report Cohen's d. The outcomes are easily interpretable as they are.

Reviewer #3: Thank you for giving us the opportunity to comment on this interesting submission, and we congratulate the authors on the effort that has been put in. 

We have a number of thoughts:

1) While we know how many centres were recruited we were not sure exactly how many GP's in each centre were taking part? Was this a consistently distributed number GPS or was there over-representation with more GPs at some centres? There was also a potentially important imbalance at baseline for DDDs.

2) We struggle to gauge the clinical significance of the findings in this paper. Although the final paragraph of the manuscript suggest that small effects could have high impact in avoiding false fractures or cognitive impairment, we believe that this trial does not provide any evidence on this. This trial only measures change in prescription, and not clinical outcomes.

3) Ultimately, it will be a health economic evaluation that is needed to judge if this sort of intervention is worth pursuing or not. Also, I am guessing that a subsequent manuscript will evaluate whether Primary Care are prepared to take on such an intervention.

Peer review: YK Loke and Navena Navaneetharaja

[LINK]

---

## [Decision Letter · Decision Letter 2]

1 Feb 2022

Dear Dr. Leiva,

Thank you very much for submitting your manuscript "Effectiveness of a multicomponent intervention consisting of education and feedback on reducing benzodiazepine prescriptions by general practitioners: BENZORED hybrid type I cluster randomized controlled trial" (PMEDICINE-D-21-03642R2) for consideration at PLOS Medicine. 

Your revised paper was evaluated by a senior editor and discussed among all the editors here. It was also discussed with an academic editor with relevant expertise, and sent to two of the original reviewers, including a statistical reviewer. The reviews are appended at the bottom of this email and any accompanying reviewer attachments can be seen via the link below:

[LINK]

In light of the remaining points of Reviewer 2, we cannot accept the manuscript for publication in the journal in its current form, but we would like to consider another revised version that addresses the reviewer's and editors' comments. Obviously we cannot make any decision about publication until we have seen the revised manuscript and your response, and we may seek re-review by one or more of the reviewers. 

We expect to receive your revised manuscript by Feb 22 2022 11:59PM. Please email us (plosmedicine@plos.org) if you have any questions or concerns.

We look forward to receiving your revised manuscript. 

Sincerely,

Caitlin Moyer, Ph.D.

Associate Editor

PLOS Medicine

plosmedicine.org

1. Data availability statement: Thank you for providing the link to access the dataset. Please revise the statement to “All data are available from the Zenodo repository (https://doi.org/10.5281/zenodo.5607352).” or similar. There may be a typo in the DOI provided, please check if this should be: https://doi.org/10.5281/zenodo.5607352

Please also check that no identifying information (GP, PHC name) are included in the file.

2. Abstract: Please structure your abstract using the PLOS Medicine headings (Background, Methods and Findings, Conclusions). Please incorporate the sentence describing the study objective as the final sentence Background.

3. Abstract: Methods and Findings: Please mention that analyses were intention to treat.

4. Abstract: Methods and Findings: Line 121-122: Please mention the characteristics for which the two groups were not apparently balanced.

5. Abstract: Methods and Findings: In the last sentence of the Abstract Methods and Findings section, please describe the main limitation(s) of the study's methodology.

6. Throughout: Please place the in-text citations within square brackets before the sentence punctuation, for example [1].

7. Throughout: Please carefully check the text for typos/grammatical errors.

8. Methods: Line 239: Please include a copy of the original study protocol as a supporting information file, and please refer to it here (for example, S1 Protocol).

9. Methods: Line 244: Please include the information regarding participant (PHC) consent and include the information that the ethical review boards waived the requirement for GP consent.

10. Methods: Line 342: Please reference the analysis plan in the Supporting Information files here.

11. Methods: Line 344-345: Please revise the CONSORT statement to: "These results are reported as per the Consolidated Standards of Reporting Trials (CONSORT) guideline extension for cluster randomized trials (S1 CONSORT Checklist)."

12. Methods: Line 362: Please provide a copy of the questionnaire as a supporting information file.

13. Results: Line 401: Please summarize the key findings from the additional analyses of effects by health district.

14. Discussion: Line 455-456: Please provide references and clarify this statement: “Prescribing BZDs in patients aged 65 years or more is considered inappropriate and should be avoided.”

15. Checklists: Thank you for including the 3 reporting checklists. Please revise the checklists, using section and paragraph numbers to refer to locations within the text (e.g. Methods, paragraph 1). Please do not refer to page numbers.

16. Supporting information Table 1: Please provide a legend for this table.

17. Analysis Plan: Please note at what point during the study the amendment to the analysis plan was made.

Comments from the reviewers:

Reviewer #2: Alex McConnachie, Statistical Review

I thank Vicens and colleagues for their consideration of my original points.

On the matter of the meaning of ITT, I think we can agree to disagree. I accept that I may be in the minority in my opinion, even if I think I am right!

However, this got me thinking about the analysis. By imputing the missing outcomes for GPs who moved out of the PHC area, the analysis is estimating the effect of the intervention under the scenario that all GPs remained within their original PHC. I am not sure this is realistic.

Also, in the analysis, it appears that only those who expressed an interest in the study prior to randomisation are included. Therefore, the analysis is estimating the impact of the intervention in a subgroup of GPs, rather than the overall impact in the population of all GPs. If implemented, some GPs might decline the intervention, and these GPs are not being included in the analysis.

Since randomisation took place at the PHC level, the fairest analysis, under ITT, might be to include all GPs within each PHC, regardless of whether they received the intervention, since the randomisation resulted in the intervention being available in some PHCs but not others. Since outcomes were collected from electronic records, would it have been possible to collect complete outcome data for all GPs in each PHC at baseline and at follow-up, removing the need for imputation of missing data?

I suppose one problem with this approach would be that for those GPs who move into an area during the study, there would be no baseline value for the outcome variables, making the analysis at GP level more difficult.

All this being said, I think that what the authors have done is very good. Both the complete case and multiple imputation results are presented, and are similar. Given that GPs who were unwilling to take part were excluded prior to randomisation, what are the implications in terms of the population-level effect of the intervention? Did these GPs, in intervention PHCs, have access to the intervention? If so, did they receive it, or did they tend to decline?

The method of estimating the marginal mean difference in DDDs from a Poisson model is fine, but should be mentioned in the statistical methods section. 

Finally, I still have a problem with the text "The Spanish Medicine Agency (AEMPS) reported an average of 87.6 DDDs per 1000 inhabitants-per-day during 2018. Thus, approximately 9% of the population of Spain is receiving on average daily equivalent of 10mg of diazepam." I do not believe that it is possible to deduce the percentage of the population who are receiving a medication, based on the average DDDs per 1000 participants. Approximately 90 DDDs per 1000 population could represent 9% of the population receiving an average of 1 DDD each, or it could represent 18% of the population receiving an average of 0.5 DDDs each. There is no way to tell.

Reviewer #3: Thank you for revising the manuscript. I don't have any further points to raise.

[LINK]

---

## [Decision Letter · Decision Letter 3]

16 Mar 2022

Dear Dr. Leiva,

Thank you very much for re-submitting your manuscript "Effectiveness of a multicomponent intervention consisting of education and feedback on reducing benzodiazepine prescriptions by general practitioners: BENZORED hybrid type I cluster randomized controlled trial" (PMEDICINE-D-21-03642R3) for review by PLOS Medicine.

I have discussed the paper with my colleagues and the academic editor and it was also seen again by one of the reviewers. I am pleased to say that provided the remaining editorial and production issues are dealt with we are planning to accept the paper for publication in the journal.

[LINK]

We look forward to receiving the revised manuscript by Mar 23 2022 11:59PM.   

Sincerely,

Caitlin Moyer, Ph.D.

Associate Editor 

PLOS Medicine

plosmedicine.org

Requests from Editors:

From the academic editor:

-Abstract: Methods and Findings Line 125: As the primary outcome is reported as absolute difference, please similarly report the secondary outcome as absolute rather than relative difference: “The intervention group also had a smaller number of adjusted relative long-term users overall (-3,6% (95%CI: 1.85,8.88, p>0,001) and a smaller number of adjusted relative long-term users over age 65 years (.- 3.5% (95%CI: 1.19, 5.71 p=0,004)”

-Methods: Study population: Line 248: I think it might be helpful to insert a brief few sentences on how PHC is organised in Spain (regions, districts, PHCs, GPs).

-Methods: Line 305: The second secondary outcome measure is perhaps better considered a sub-group analysis (people over 65 years who are long term BZD users). Was there any heterogeneity of effect between those >65 years and those <=65 years? It looks like there was also a pre-specified sub-group analysis by district (e.g. described at line 339-341 and reported at line 406). It would be good to have a sub-heading on pre-specified sub-group analyses and ensure all are reported in the results.

-Results, Table 2: Please include a footnote to describe what covariates were included in the adjusted model. It is difficult to work out from the main manuscript text.

-Results: I am a little surprised that with the large actual ICC (0.3) relative to the sample size calculations (0.05) that the study was adequately powered to show a difference – the effect size is smaller and the number of general practices is only a little higher compared to the original calculations. But it is hard to calculate the actual study power based on the data providers (e.g the patient cluster sizes at the PHC level are not given). Could this be checked?

-Results: Also related to the point above it would be helpful to understand a little more what is happening at the PHC level to cause such a high level of clustering? This is related to my comment above on PHC structure in Spain. Most of the discussion focuses on the level of GPs and while I can understand within general practice clustering I am a little unclear what is driving PHC level clustering.

Other editorial points:

1. Title: Please revise to: “Evaluation of a multicomponent intervention consisting of education and feedback to reduce benzodiazepine prescriptions by general practitioners: The BENZORED hybrid type I cluster randomized controlled trial” and please update this in the manuscript submission system as well as the text of the manuscript.

2. Data availability statement: We suggest revising to: “The de-identified data underlying the study’s findings are publicly available in the Zenodo repository: https://doi.org/10.5281/zenodo.5970482”

3. Throughout: Please edit for minor typos and grammatical errors throughout.

4. Abstract: Line 112-113: “GPs were not blinded to the allocation.” Please also mention who was blinded to the allocation.

5. Abstract: Line 114: Please clarify if “tapering” rather than “withdrawal” would be preferable here.

6. Abstract: Line 115: We suggest revising to: “...with monthly BZD prescription feedback and access to a support webpage.”

7. Abstract: Line 116-117: Please clarify the number of GPs (and PHCs) lost to follow up in each group.

8. Abstract: Line 119: Please define “long term users” in this sentence.

9. Abstract: Line 131: Please provide a definition of “physician doctor” as this may not be a universally understood term.

10. Author Summary: Line 165-167: We suggest revising to: “... successfully reduced the proportion of long-term users, which could potentially reduce adverse outcomes related to long-term use.” or similar.

11. Line 172: Please change “Background” to “Introduction” as the heading for this section.

12. In-text citations: Throughout, please include a space between the reference bracket and the preceding word, for example [1]. Please also check that all reference brackets are placed before the sentence punctuation, where applicable.

13. Methods: Please clarify the nature of the informed consent provided by GPs at PHCs included in the study.“Only PHCs in which at least two-thirds of the GPs agreed and provided written informed consent to participate were included. Informed consent was waived by the Balearic Islands and the l'IDIAP Jordi Gol Ethical Committee of Clinical Research because the analysis used anonymous administrative data. The requirement for informed consent was not waived by the Valencian Primary Care Ethics Committee, and therefore, from this region, only PHCs from which all GPs agreed and provided written informed consent were included.” or similar, if this is accurate.

14. Methods: Line 294-295: “This web site included videos, descriptions of practical cases, and an information leaflet for patients about BZDs, Z-drugs, and sleep hygiene (http://benzored.es).” Please note that the link provided does not seem to be functional, please include an accessible link.

15. Methods: Line 300-301: “Total number of dispensed DDD from the N05BA, N05CD, and N05CF groups of the Anatomical Therapeutic Chemical (ATC)Classification System code…” please clarify what prescription drugs are included in these codes.

16. Results: Line 378-380: We suggest clarifying to: “This study was conducted in three Spanish health districts, which included 90 PHCs (58 in the Balearic Islands, 20 in the Tarragona-Reus district, and 12 in the Arnau de Vilanova-Llíria district) and 867 eligible GPs (Figure 1).”

17. Results: Line 391-392: Please report the baseline DDD per 1000 inhabitants per day for the intervention and control groups in the text.

18. Results: Line 418-425: Please mention in the text what factors were adjusted for in these analyses.

19. Results: Line 417: For secondary outcomes, it would be helpful to report the absolute differences, as mentioned by the Academic Editor (please see the comment on the Abstract).

20. Results: Line 427-436: Please present data on these results, e.g. percentage of respondents who reported on each item of the questionnaire. Please consider adding a table with complete results for these outcomes.

21. Discussion: Line 459-462: “In contrast, other authors found that educational visits with GPs regarding BZD prescriptions were not effective in reducing the number of prescriptions[52,53]. Pimlott et al. and Holm also found that an educational intervention and feedback to GPs was not effective[54,55].” Some additional discussion on why these educational/feedback interventions may have been less effective would be helpful to provide context.

22. Discussion: Line 468-470: Please revise if it might be more accurate to interpret this as: “We found that our intervention led to a 3.5% reduction in the proportion of long-term BZDs users who were aged 65 years or more.”

23. Discussion: Line 472-473: We suggest revising to “A previous study of a stepped-dose reduction of BZDs, with written instructions delivered by GPs to long-term users…” or similar.

24. Discussion: Line 477-479: Please clarify here and throughout, as the secondary analysis revealed a reduction in proportion of long-term BZD users, rather than a reduction in BZD use by long term BZD users. We suggest changing to: “These results differ from our finding that the intervention implemented by GPs provided only a modest reduction in the percentage of long-term BZD users.”

25. Discussion: Line 503-505: Please include slightly more context here, briefly discussing how these local policy differences could have influenced your results. Please also include relevant citations: “For example, the Balearic Islands and Tarragona have local policies regarding BZD prescribing that include indicators and incentives.”

26. Discussion: Line 506-507: We suggest revising to: “This suggests the results may be applicable in different clinical settings.”

27. Line 538: Please remove the “Author contributions” section from the main manuscript text, and ensure all information is completely entered into the relevant sections of the manuscript submission system.

28. Line 553: Please remove the “Funding statement” section from the main manuscript text, and ensure all information is completely entered into the relevant sections of the manuscript submission system.

29. Line 564: Please remove the “Competing interests statement” section from the main manuscript text, and ensure all information is completely entered into the relevant sections of the manuscript submission system.

30. Line 566: Please remove the Ethics and Dissemination section from the end of the manuscript, and ensure the information is included in the appropriate location of the Methods section (this information could be added to the “Study Design” sub-section, or as a separate sub-section). The sentence “All data are available upon reasonable request.” can be removed as the information on accessing the publicly available data is included in the Data Availability statement of the manuscript submission system.

31. References: Please check the formatting of each reference in the list. Please use the "Vancouver" style for reference formatting, and see our website for other reference guidelines https://journals.plos.org/plosmedicine/s/submission-guidelines#loc-references

For example:

Reference 4: The journal title should be “BMJ”

Reference 6: The journal title is missing.

Reference 9: This reference is incomplete.

Reference 27: This reference is incomplete.

Reference 30: Please check the journal title.

32. Table 1: Please define “GP Specialty Training” as this was not mentioned in the Methods (please describe assessment of GP characteristics in the Methods).

33. Table 2: In the legend, it would be helpful to define that the table reports on the percentage of all patients who are long term BZD users (and how this is defined), and the percentage of long-term BZD users who are older than 65 years. Please mention the factors adjusted for in the analyses in the legend (i.e. please briefly describe the unadjusted, adjusted, and estimated ITT analyses).

34. Supporting Information Table 1: Please remove the duplicate file (the file without p values).

35. TIDieR, StaRI, and CONSORT Checklists: Please remove the older versions of these files.

36. S1 Implementation Fidelity Questionnaire: Please include a copy of the questionnaire as a supporting information file.

37. Study protocol: Thank you for including the original copy of the study protocol (“renamed_11888”). Please include a version in English, and we suggest renaming the file (e.g. S1 Protocol, or similar).

Comments from Reviewers:

Reviewer #2: Alex McConnachie, Statistical review

I thank the authors once again for their responses. The section about DDDs and percentage of users has been removed, which was my only real objection. Whilst I feel there are other ways that the analysis could have been done, I am happy with the analyses as presented, and I have no further comments to make.

[LINK]

---

## [Editor Report · Decision Letter 4]

7 Apr 2022

Dear Dr Leiva, 

On behalf of my colleagues and the Academic Editor, Dr Peiris, I am pleased to inform you that we have agreed to publish your manuscript "Evaluation of a multicomponent intervention consisting of education and feedback to reduce benzodiazepine prescriptions by general practitioners: The BENZORED hybrid type 1 cluster randomized controlled trial." (PMEDICINE-D-21-03642R4) in PLOS Medicine.

Prior to final acceptance, please address some minor points:

At line 123, please adapt the text to "forty"; and use "follow-up" here and at any other instances;

At line 162, please spell out "DDD" at first use in the Author Summary; 

Please remove the academic editor's name from reference 5; 

Please remove the duplicated text from reference 9; and 

Please correct reference 34, removing the Orcid and the author name following the title. 

PRESS

Sincerely, 

Richard Turner PhD, for Caitlin Moyer, Ph.D. 

rturner@plos.org